# Three-Dimensional Bioprinting Applications for Bone Tissue Engineering

**DOI:** 10.3390/cells12091230

**Published:** 2023-04-24

**Authors:** Jamie A. Maresca, Derek C. DeMel, Grayson A. Wagner, Colin Haase, John P. Geibel

**Affiliations:** 1The John B. Pierce Laboratory, University of New Haven, New Haven, CT 06519, USA; 2Yale School of Engineering & Applied Science, Yale University, New Haven, CT 06519, USA; 3Department of Surgery, School of Medicine, Yale University, New Haven, CT 06519, USA

**Keywords:** hydrogels, bioink, osteoblast, mesenchymal stem cell, bone replacement, scaffold

## Abstract

The skeletal system is a key support structure within the body. Bones have unique abilities to grow and regenerate after injury. Some injuries or degeneration of the tissues cannot rebound and must be repaired by the implantation of foreign objects following injury or disease. This process is invasive and does not always improve the quality of life of the patient. New techniques have arisen that can improve bone replacement or repair. 3D bioprinting employs a printer capable of printing biological materials in multiple directions. 3D bioprinting potentially requires multiple steps and additional support structures, which may include the use of hydrogels for scaffolding. In this review, we discuss normal bone physiology and pathophysiology and how bioprinting can be adapted to further the field of bone tissue engineering.

## 1. Introduction

Three-dimensional (3D) bioprinting describes the use of 3D additive manufacturing techniques aimed to integrate biological materials, such as cells, growth factors, and other biochemicals and biomaterials, into a multi-layer composite using high-precision printing technologies that can mimic the structures of target tissues [1,2]. This allows for the reproducible automated production of functional living tissues. Bioprinting has garnered continued interest in the healthcare community for its ability to generate complex biological structures such as organs, blood vessels, and bones, leading to extraordinary advances in regenerative medicine [3]. Bioprinting technologies have allowed medical scientists and biomedical engineers to research new treatment modalities in cases where substantial portions of bone and tissue are removed or destroyed due to a traumatic injury or chemical or radiation exposure. The ability to repair bone using bioprinting techniques has especially gained attention due to its potential as a “game-changing” treatment to combat the limited availability of organ donors in the United States [3]. The complications surrounding “bone grafting”, the name given to procedures that use the transplanted bone to repair and rebuild diseased or damaged bones, have also garnished a large amount of attention from the biomedical community due to potentially reducing the risk of immunological responses. Some of the highlighted difficulties are around the problems of absorption and reabsorption of the graft material and the risk of infection associated with the procedures. To address these problems, different mixtures of cells, scaffold materials, and bioink materials have been proposed and are being studied further to produce mechanically strong, biocompatible, and durable graft properties capable of achieving the goals required for bone repair [1,4].

In this review, we will examine the currently deployed bioprinting technologies along with the various substrates implemented. Additionally, bioprinting technology gained attention recently as a potentially viable solution for bone remodeling and repair. Due to the printer’s ability to extrude a novel composition that can regenerate or repair biological tissues such as bone, it has been found as a new candidate for generating bone grafts, and eventually full-length bone replacements. There are several different techniques that can facilitate 3D bioprinting, which will be introduced in the following subsections.

### 1.1. 3D Bioprinting Processes and Approaches

#### 1.1.1. Bioink

What is bioink? Bioink is any material composed of cells that are suspended in a media containing additional material for cell growth, continued cell structure, and ink viscosity to allow for printing. The formulation and subsequent extrusion of a bioink are undoubtedly one of the most difficult characteristics of the 3D bioprinting process. A well-formulated ink should create the desired substrate to eventually develop structure while providing the printed cells with nutrients to survive. It is also important when developing bioink to assure that it can facilitate oxygen transport to the growing cells to allow for adhesion to the substrate and must be free of cytotoxic compounds to provide the optimal growing environment [5]. Along with these aspects, the “ink” must be able to absorb O_2_ and nutrients and extrude CO_2_ and waste products, thereby preventing changes in pH, along with fluid and electrolyte composition in the microenvironment, mimicking the native tissue in the normal physiological environment.

#### 1.1.2. Extrusion-Based Bioprinting

Extrusion-based 3D bioprinting is the most common method employed within the field of 3D bioprinting at the present time. This method employs the release of material through a small needle or needle-like nozzle attached to a reservoir (typically a syringe of varying volumes) to generate a desired shape by utilization of a bioink, which suspends the printed cells in a media containing growth factors, cells, and other biochemicals essential to the printing process and that aids in cell viability before and after printing (Figure 1). Inkjet printing is a low-cost, high-efficiency, and moderately precise printing method, basic in its operation, where microdroplets of a liquid bioink are extruded onto specific areas of a substrate [6]. These printers can also be optimized using computer-aided design (CAD) programs to allow for highly precise printing, facilitating the ability to print highly specific shapes with a set amount of bioink material.

#### 1.1.3. Laser-Assisted 3D Bioprinting

Laser-assisted bioprinting is one of the newest methods of 3D bioprinting. It involves the layering of a ribbon of glass oriented parallel to the collecting substrate, a ribbon of laser-focusing material such as gold, and a sheet of hydrogel, bioink, and biomaterials. The biomaterial is placed above a substrate, the glass and gold sheet are layered on top of this biomaterial, and once a laser pulse strikes a point on the gold ribbon, it creates a miniature burst of vapor that drives the biomaterial onto the substrate (Figure 1). Laser printing allows for the construction of more complex biological structures and complexes using laser technology, which removes any physical stress upon the cells suspended in the bioink [6,7]. This process allows for a higher resolution when printing, generating more precise shapes to model the design of the organ that is being repaired or potentially replaced. Laser-assisted techniques also lack a nozzle, meaning that the clogging issues found commonly in inkjet and extrusion printers are completely absent. However, laser-assisted bioprinting is expensive, making it an unfavorable candidate for daily research without spending a lot of funding. Except for the cost, laser-assisted techniques are vastly superior to inkjet and extrusion printers, and with more printers being deployed in the future, the costs may be reduced. While printing at a slightly slower rate, laser printers can extrude bioink with much more dense cell concentrations of up to 10^8^ cells per mL, equating to around a single cell per microdroplet [6,7]. In addition to their ability to print a larger volume of cells, laser printers have the highest cell viability of all the printing techniques, with around 95% viability using this method.

#### 1.1.4. Hydrogels as Scaffolding for 3D Bioprinting

Recent exploration in bioprinting technologies has focused on scaffold-free printing by further optimizing bioink used to provide the initial structural and mechanical support for printed cells. Hydrogels are an essential piece of 3D bioprinting as they provide a 3D structure and proper biochemical environment that can effectively simulate native bone tissue [8]. They are crosslinked hydrophilic polymers capable of absorbing substantial amounts of water or bioink without dissolving in them [9]. Hydrogel is widely used with many 3D bioprinting techniques, as it creates an adequate biological environment for cells to grow. The requirements for a suitable bioink are truly extensive due to biocompatibility issues across a wide range of cells and other tissue-specific properties, so hydrogels are now generally settled upon as the optimal substrate for bioprinting [8,10]. Though the dynamic biochemical characteristics of living tissue differ from the invariable conditions of hydrogels, it is part of this review to assess the development of hydrogels and discuss some of the modifications aimed at accommodating the potential flaws of the presently used hydrogel systems. The type of hydrogels used in bioprinting varies widely depending on the printing technique [10]. In this review, we will be covering hydrogels commonly used for bone applications, such as alginates, a low-cost, algae-derived material. It is a naturally occurring polysaccharide composed of guluronic and mannuronic acids. Alginate, on top of being widely available, is highly biocompatible with humans and forms a hydrogel under appropriate conditions. This review will discuss hydrogels used in 3D bioprinting in greater detail.

## 2. Bone Remodeling and Current Strategies for Repair

### 2.1. Bone Biology

Bone is a dynamic tissue undergoing constant remodeling [11,12]. Bone remodeling is the biological process where new bone tissue continuously replaces old bone. It is essential for allowing bones to maintain their structural integrity and adapt their skeletal architecture in response to changes in mechanical stress [12,13]. There are three types of bone cells involved in the resorption, reversal, and formation phases of the bone remodeling process: osteoblasts, osteocytes, and osteoclasts [11,14,15,16]. Osteoblasts are the builder cells that create new bone tissue and express proteins involved in bone remodeling and mineralization [15,17]. They are differentiated from mesenchymal stem cells and become osteocytes as they mature and become embedded into their own extracellular matrix [14]. Osteocytes continue to generate the organic tissue and further construct the bone network [15,17]. In contrast, osteoclasts are the repairers; they are bone-resorbing cells that erode the bone matrix using acids and enzymes to complete the bone remodeling process [14].

Beyond their own bone cells, bones also contain several other organic and inorganic compounds, including mineralized osseous tissue, bone marrow, endosteum and periosteum, nerves, blood vessels, and cartilage [14]. For example, all long bones, including those in the shoulder, arms, and legs, have three primary regions: the epiphysis, the diaphysis, the epiphysis, and the epiphyseal plates [14], as shown in Figure 2. The former makes up the long portion of these bones and consists primarily of cortical (compact) bone that makes up about 80% of its total mass [14]. Contrastingly, the epiphysis contains primarily cancellous (spongy) bone and red bone marrow. The epiphyseal (growth) plate is active only during the bone growth process and promotes bone lengthening. Once the bone has finished growing, the epiphyseal plate ossifies [14].

### 2.2. Fracture Types and Repair Methods

A bone fracture (or osteotomy) is any breakage in the anatomic continuity of the bone such that it loses its mechanical stability [18,19]. There are several different ways in which bones can be fractured. These include open fractures, where part of the bone breaks through the skin; closed or simple fractures, where the bone does not pierce the skin; partial (incomplete) fractures; complete fractures, where the bones fragment into two or more pieces; stable fractures, where the ends of the broken bone are still aligned; and displaced fractures, where a gap is left between the ends of the broken bone [20]. These broad categories also encompass more specific injuries, including transverse fractures, oblique fractures, segmental fractures, and comminuted fractures, which all differ in the geometry of the breakages, as shown in Figure 3 [19,21]. Depending on the location of the fracture and its type, different approaches are taken for repair. External and internal fixation are the current standards for stabilizing bone to initiate osteogenesis; however, 3D bioprinting technologies may offer even more possibilities.

External fixation is used for complex fractures and corrective osteotomies such as limb-lengthening procedures [20,21]. It involves the surgical implantation of devices such as pins and screws that are later removed. In contrast, internal fixation is used for debilitating conditions such as total joint fractures. It involves a more extensive surgical approach with the implantation of prosthetics and other permanent devices to allow joints and soft tissues to maintain their normal functions as the bones repair themselves [20,22,23]. Internal fixation can be further separated into two categories: Open Reduction and Internal Fixation (ORIF) and total joint arthroplasty [24,25,26,27]. ORIF involves permanently implanting screws and plates to hold the fragmented bones in place. It is the preferred intervention for joint fractures such as the hip, where only a partial component of the joint requires removal (i.e., removal of the acetabular socket but not the femoral head for a hip injury) [26,28]. In contrast, total joint arthroplasty (or joint replacement surgery) involves removing part or all of a diseased joint and replacing it with prosthetics made of metal, plastic, or ceramic to restore the normal movement and function of the joint [29,30]. It is chosen for repairing high-energy fractures such as those from car accidents, high falls in the elderly population, or debilitating diseases like osteoarthritis [29,30].

### 2.3. Limitations of Existing Treatment Methods

Total joint arthroplasty is highly regarded as the gold standard intervention for joints that became osteoarthritic, yet it is not invulnerable to serious complications. The ability to promote osteogenesis is dependent on ample fixation of the devices following the surgery and on the mechanical loads placed on the implants during normal movement [31,32]. Nonetheless, a striking 5% of arthroplasties result in failure due to inflammation at the surgical site, post-surgical infections, and loosening of the implant through wear [27,32,33]. Prosthetics that lack adequate initial fixation experience micromotion during normal skeletal movement. Over time, this movement can accumulate into a gradual implant migration [31,32]. Additionally, normal movement may also cause the metals, plastics, and ceramics used in traditional implants to release particle debris as they rub against surfaces. The particle debris may then lead to infection, erosive bone resorption, and tissue swelling [32,33]. All of these factors increase the likelihood that the implant may warrant removal or replacement over time. In fact, a study by Mittal et al. determined that surgical revisions, including implant removal, are one of the greatest limiting factors of traditional implants due to their high frequency [27,34,35].

Bone and soft tissue necrosis are other phenomena that occur as a consequence of trauma. Direct manipulation of the bone by the surgeon during internal fixation procedures—such as during the preparation for device implantation—results in iatrogenic bone necrosis [23]. It can develop from active processes such as bone reaming and passively from contact between the implant and periosteal and/or endosteal surfaces, which carry the blood supply to the bone [23]. The implant inhibits the amount of blood that can reach or leave the bone, thus resulting in soft tissue necrosis [23]. Furthermore, the contact between the implant and necrotic bone may impair the bone-remodeling process and impede infection resistance. This is of concern as the inhibition of bone healing may instead facilitate bone refracture due to tensile loads created by the implants and, therefore, may eventually require the implant to be removed [23]. These limitations reveal a need for alternative methods of treating bone fractures and disease. The development of novel biomaterials has been applied to address the replacement or augmentation of bone tissue. Osteogenic scaffolds, grafts, and implants show potential for clinical use to address these issues.

### 2.4. Biomaterials in Implants and Bone Scaffold Development

There are many instances where a diseased or injured bone is unable to repair itself fully using mechanical fixtures alone, thus resulting in a gap or nonunion between the segments [14]. Bone grafts are a common tool used for treating this and other bone defects [14]. They can be derived from autogenous cancellous, cortico-cancellous, or cortical bone. Autologous bone grafts are the current standard intervention where bone from the host is removed from elsewhere in the body (typically from the pelvis or iliac crest) [14]. However, they carry major drawbacks, including high pain, inflammation, risk of infection and disease transmission, and morbidity at the donor sight as high as 30% [14]. Allografts, tissues derived from other humans, are another alternative material. To be considered, the allograft bone tissue needs to be biologically similar to the patient’s own tissue to minimize immunological incompatibility [14,16]. Nonetheless, orthopedic allografts introduce additional concerns such as host immune rejection post-implantation and a disease transmission rate as high as 13% [14].

To address these concerns, several synthetic bone scaffolds have been engineered over recent years [14]. Popular commercially available bone graft substitutes include demineralized bone matrix, collagen and mineral composites, and calcium sulfates. These synthetic tissues are designed to mimic the porous architecture of the host environment and provide comparable mechanical strength to promote the desired bone cell migration and osteogenic differentiation [14,34]. They are also designed to biodegrade in a controlled manner over time as the bone repairs itself [14]. A well-designed scaffold should maintain its mechanical properties for 1–3 months post-implantation and degrade after 12 to 18 months so that the material can be absorbed through metabolic pathways and not impede bone tissue growth [14]. Other key advantages of synthetic scaffolds include their widespread availability, reduced risk of infection, and total evasion of transmitting any diseases to the recipient. Nonetheless, their widespread use is gate kept by the fact that the patient must wait for the bone graft to grow in vitro before it can be implanted [14].

Many synthetic materials may be utilized as long as they satisfy the criteria of being biocompatible, biodegradable, and capable of withstanding the loads and other mechanical forces that bones receive. Popular candidates include polyglycolide (PGA), polylactide (PLLA), and polylactide-co glycolic acid (PGLA). Though engineers are now also incorporating elastin, collagen, and hyaluronic acid to create devices with even more bone-like composition, as listed in Figure 4 [34,36]. However, these materials are still expected to initiate some form of immune response when implanted as they are foreign materials. Properties such as internal geometry (porosity and pore), outer surface microtopography, and degradation profile all impact the degree of inflammation that occurs from the foreign body response [34,36,37]. These material properties also determine how well the implant will perform and be tolerated to facilitate bone regeneration. For example, implants with pointy geometries will induce a greater inflammatory response [34]. Implants with porosities of between 80 and 90% and pore sizes of around 300 μm allow for optimal bone tissue in-growth [32,37]. Implants with larger and smoother surface areas will experience greater absorption and platelet adhesion—factors that mediate acute inflammatory responses to implanted biomaterials [34,37].

Beyond implants and bone grafts, bone cells themselves are being used as the basis for several successful 2D and 3D nanofabrication techniques for bone scaffolds [16]. Scaffolds are a critical component of bone tissue engineering [14,16]. They are three-dimensional structures that replicate the critical properties of natural extracellular matrix and provide the framework for cells to latch onto to produce bone tissue [16]. Recall that the primary goal in bone scaffold design, much like that of implant design, is to mimic the micro and nano-scale characteristics of bone [14]. As stated earlier, porosity and surface texture directly influence the ability of bone cells to grow and for extracellular matrices to be produced with sufficient structural integrity [14].

This brings us to the use of 3D bioprinting as an alternative method for producing synthetic bone tissue. Three-dimensional bioprinting offers even more applications for bone repair. It is a fabrication technology used to construct complex and functional three-dimensional living tissues through the deposition of cells in a bioink [39,40]. Three-dimensional bioprinting technology as a whole is still in its infancy; however, it is promising in developing implantable bone tissues using a patient’s own cells to generate customizable, implantable bone tissues without the concerns of the current implant and synthetic biomaterials [40]. The focus of much 3D bioprinting research is on the development of scaffolds that enhance osteogenesis and can later be utilized by a surgeon to repair defects in a patient’s own bones [41]. Importantly, these scaffolds can be customizable to the patient’s individual anatomy and, therefore, any specific bone defects or fractures with appropriate morphology, mechanical strength, and chemical properties [41].

## 3. Bioprinting and Bone

Three-dimensional bioprinting technology can be used as an additional potential solution to heal bony defects and create bone tissue substitutes. As previously discussed, extrusion, inkjet, and light-based 3D bioprinting have been shown to create tissue with osteoinductive properties [42]. Bioprinted bone grafts could be used as a substitute for allografts, which would reduce the risk of graft rejection and chronic inflammation [42]. Alternative methods for creating bone tissue include 3D-printed scaffolds that are then seeded with cells, but there are several disadvantages to this, such as non-uniformity of cell distribution [43]. However, bioprinting has the advantage of an even cellular distribution and the ability to create detailed shapes [42]. Furthermore, bioprinting also allows for a greater level of precision and control over structural components such as porosity [42,44].

### Bioink Development

Developing a suitable bioink for printing remains a common challenge as there are many requirements it must satisfy. When extruding, it must be able to hold a stable conformation and deliver cellular components with minimum damage, as well as mimic the native cellular environment [42,45]. The bioink must have strong mechanical properties and biocompatibility while still being able to extrude easily into a printed construct [42]. In the case of printing bone tissue, stiffness is particularly important for cell attachment and proliferation, which is necessary to promote the differentiation of human mesenchymal stem cells (hMSCs) into osteogenic cells [45]. It is logical to use bioinks that match the native environment of the goal tissue, and in the case of printing bone, the ECM environment produced by the bioink should be close to 25–40 kPa in stiffness [45]. In addition to mechanical strength, bioprinted constructs that are biodegradable allow for efficient remodeling and integration into the implant site. A robust and biomimetic bioink should be chosen with an optimal degradation rate [45]. The key to successfully bioprinting tissue is finding a bioink with both high printability and biocompatibility [43]. For example, a recent study found that using nanoengineered-ionic-covalent-entanglement (NICE) bioinks that are enzymatically degradable provided mechanical reinforcement and osteogenic differentiation when used for osteogenic tissue bioprinting [45].

Bioinks are typically hydrogels, meaning there is high water content. Hydrogels are known to be compatible with bioprinting and can mimic the ECM [46]. The chosen bioink must be able to print in a liquid form and then become a stable gel structure [42]. Three common hydrogels chosen in bone tissue applications include alginate, gelatin, and collagen [44,47]. Alginate is a popular polymer used in bioprinting due to its natural biocompatibility and biomimetic properties [43]. It is derived from algae and has a unique ability to control gelation, making it ideal for bioprinting applications [43]. Alginate has been shown to safely encapsulate cells and provide a safe environment resulting in high cell viability after printing [43]. However, alginate has relatively weak mechanical properties, so it is less commonly implanted in bioprinting bone tissue [43]. Another common example is gelatin methacrylol, or GelMA, which is cross-linkable under UV light and mimics the extracellular matrix environment [45,46]. GelMA is known to have high as well as desirable mechanical properties [46]. The stiffness and behavior of GelMA are easily tunable, and previous studies have found that GelMA works well in conjunction with bone-marrow-derived mesenchymal stem cells (BMSCs). However, GelMA is known to have poor printability [45]. Collagen is used as well due to its greater biocompatibility, although there tend to be issues with low viscosity and lack of stability [44]. There have been a wide variety of bioinks used in bone bioprinting research, including combinations of these three primary hydrogels and additional components. A recent study by Sawyer et al. found that a mix of gelatin, alginate, and collagen with encapsulated human mesenchymal stromal cells (hMSCs) could be optimized for bioprinting [47].

Not only must a bioink have high printability, but it must also maintain cell viability during and after printing. Just as in native bone tissue, there are many different cell types that should be included in a bioprinted bone construct. Primarily osteogenic and angiogenic cells must be considered [44]. A critical component of a successful bone graft is osteoinduction, which requires mesenchymal stem cells (MSCs) to differentiate into osteoblasts, typically with the help of bone morphogenetic proteins (BMPs) [42]. It is possible for some materials to naturally induce osteogenic differentiation, such as calcium phosphate ceramics [42,43]. Cells typically used include bone marrow stromal cells, endothelial cells, and induced pluripotent stem cells [44].

In addition to the required cellular components, the incorporation of inorganic components and growth factors is suggested to assist in osteoinductivity [44,46]. The most common growth factors used include BMP-2 and vascular endothelial growth factor [44]. There has also been work performed on autologous growth factors used by the patient themselves to not only reduce the inflammatory response but also promote construct integration [46,48]. However, the current gold standard is to use autologous bone particles to assist osteogenesis, which could be incorporated into a bioink. A recent study found that the addition of autologous bone particles in a GelMA-based bioink benefits the mechanical properties of the construct as well as improved bioink functionality [46]. The addition of osteoconductive materials, such as hydroxyapatite (HAp), as well as bioactive glass, will increase osteogenic differentiation [44].

There are many challenges to overcome before 3D-bioprinted bone tissue can be reliably applied to the clinical setting. The stability and size of current constructs have been limited [44]. Previous experiments using extrusion-based bioprinting have reached a size of 1–2 cm [44]. A common obstacle with all bioprinted tissues has been achieving vascularization. To maintain cell viability throughout the entire construct, there must be a vascular network to deliver nutrients. The potential applications for bioprinted bone substitutes have been explored by many, as seen in Table 1, including the repair of craniomaxillofacial defects, general fractures, and drug delivery. There has even been work conducted on the idea of bioprinting bone in situ. Di Bella et al. demonstrated the use of a “Biopen” for cartilage regeneration, which is a handheld device that utilizes coaxial bioprinting to simultaneously extrude both MSCs and a GelMA-based scaffold [49]. This can be used during surgery to directly apply cells to repair a cartilage defect (Figure 5).

## 4. Discussion

Discussed throughout this review are the various presently deployed ways of implementing 3D bioprinting that can be used with the accepted technologies and what pathophysiological and physiological aspects are required for creating a functional print. Many technological advancements have been made to the processes that go into 3D bioprinting in recent years [61]. Although there have been many advancements in the field, new and novel print technologies, such as Biological 3DJet Printing, are now being deployed, and the prospect for new and improved technologies or combinations of technologies may allow for a higher degree of success and prolonged survivability of the print in the recipient.

Current methods for 3D bioprinting include extrusion-based, laser-assisted, bioreactor, and inkjet printing [61,62]. The current 3D printers rely on the extrusion of encapsulated cells onto a scaffold or premade structure; new printer development allows for a new method called reactive jet impingement [63]. The reactive jet impingement allows for cells to be jetted at a higher density when encapsulated [63]. With a printer that is capable of extruding a high density of cells that are mixed with a form of a hydrogel, it is possible it can open doors for future developments in bone tissue engineering [63]. The new developments in 3D bioprinting bone scaffolds and advancements in hydrogel research discussed in this review can help direct research toward a better solution to bone injury surgery.

### 4.1. Bone Replacement Advances

The disease of the bone and even injury often result in the need for surgery to repair or replace damaged sections of the bone affected. The methods used in the past and most often practiced today have room for improvement by using past ideas and molding them into a superior new functioning and targeted compatibility to each patient [64,65]. Current research for extrusion-based printing involves using ceramic materials, such as zirconium oxide, which are being used in bone replacement therapies. Porous scaffolds were created with PLA filaments, the scaffolds were submerged into a mixture of compounds to create a ceramic mold around the scaffold, and thus, a stronger and more dense structure was formed [64]. The scaffolds that are created for structures to be used in bone replacement surgery can allow for higher compatibility and regeneration of vascularization with the patient over non-3D-bioprinted methods [64,65]. The bone structure itself is a porous structure, replacing segments of bone with non-porous implants can lead to complications. Through the development of the scaffold structures into a strong yet porous hexagonal shape, it will lead to a strong implant with similar force resistance as bone and high compatibility in vivo with the patient [65,66]. These advancements will improve the quality of life of patients and may allow the individual to return to previous levels of mobility and activity that occurred prior to the injury.

Three-dimensional bioprinter bioreactors have experienced new developments similar to extrusion printers and ceramic scaffolds. The use of a bioreactor printer can produce bone tissue structures using uni- and bi-directional perfusion chambers for cell culture [62]. Three-dimensional scaffolds are printed from a bioprinter, then cultured within the bioreactor to form a bone graft. Cell mixtures are added to the scaffolding to help formulate the bone structure. This method can promote the regeneration of bone structure in vivo due to cells used, mesenchymal stem cells, and the creation of a biodegradable implant that bone regeneration can replace [62,67]. Additionally, a recent paper has outlined that dental pulp stem cells (DCPS) have shown a greater differentiation potential. DCPS have shown low morbidity and neurogenic differentiation. The stem cells are extracted from wisdom teeth that were surgically removed. Thus, DCPS could be used in the tissue engineering of bone grafts due to their high differentiation potential [68]. The implantation of a bone graft structure that can be replaced with natural bone regeneration can improve the methods and surgeries typically used for bone replacement. Additionally, the utilization of highly differentiated cells, such as the DCPs, allows for further advancements in more complex grafts.

The creation of scaffolds, for 3D printing BTE, can greatly increase the quality of life of a patient post-surgery and injury from the development of ceramics or even biodegradable implants. Bone, especially long bone, needs to be able to withstand extensive pressure and force, with advancements in the 3D bioprinting of bone structures, better compatible structures can be implanted for patients suffering an injury to the bone [66]. From the gathered research, we suggest that in order to further advance the field of bone replacement, additional research needs to go into creating new scaffolds that are biodegradable allowing for the use of cells or stem cells to regenerate the damaged bone in situ.

### 4.2. Hydrogels in 3D Bioprinting and Where to Improve

In the formation of bioinks, a key component is hydrogels. Discussed in this review is the importance of hydrogels to bioinks and 3D bioprinting. Hydrogels are a key component of the bioprinting process because of their unique capability to create an artificial extracellular matrix. Collagen, gelatin, and alginate help facilitate the regeneration of bone cartilage by recreating the extracellular environment [69]. Alginate has become an industry favorite in attempting to 3D bioprint bone tissue, however, some studies show that the strength of alginate does not compare to the strength of bone tissue [69,70]. Current research on the use of alginate and other polysaccharide hydrogels shows promising results on how to formulate a stronger and more effective structure for bone tissue engineering.

Optimization of alginate printing modalities can allow for improved withstanding of pressure put on alginate when testing the capability to replace bone over time compared to bone. Different sodium alginates contain varying concentrations of G (guluronate); the changing of G concentration allows for differing properties. The alginate performed best with the highest G percentage [69]. Through exploration of the components that make up alginate, the optimized hydrogel showed the most promising results to be used in 3D bioprinting future directions. Other studies suggest the use of multiple forms of hydrogel to be added to a mixture for the best functioning hydrogel for bone tissue engineering, the use of collagen alongside the high percentage of G alginate could create the best capable extracellular environment for bone tissue growth [69,71].

Collagen is another commonly used hydrogel matrix alongside alginate. Both hydrogel components have benefits as to why they would be selected for bioink. Alginate adds mechanical strength to a 3D-bioprinted structure, while collagen mimics the extracellular matrix without adverse effects on the body [71]. The two could be combined. Alginate was dissolved in a buffer, and tyramine and other components were added in a 1:1:1 ratio with the alginate to create a hydrogel mixture. The hydrogel mixture was used to create a dynamic hydrogel that could be used as a scaffold for a print of an artificial vessel structure. The mixture created a faster, more efficient way to create a scaffold; then, cells were injected and cultured to create a vessel structure [71]. This study created a flexible vessel from the mixture and combination of the two components to hydrogel. The advancements in 3D bioprinting of BTE are rapidly being discovered. The discoveries that are made allow for revolutionary changes in all fields, especially bone defects, disease, and injury repairs. A study from early this year was able to produce a cartilaginous BTE and use it in vivo in mice to repair a bone defect. The scaffolds that were loaded with collagen were able to stimulate the deposit of new bone to the center of the defect [72]. Future studies can use the results from the two studies and their use of collagen to formulate hydrogel options that mimic bone tissue scaffold.

The issue with bone tissue engineering and the use of 3D printing for bone tissue grafts and replacement is the lack of regeneration ability of bone. A study performed was able to develop a self-healing hydrogel structure [73] that could further the research for bone tissue engineering. Konjac glucomannan polysaccharide was the hydrogel used in the study. KGM has ideal properties for use in 3D bioprinting. KGM can be cross-linked in an oxidation reaction, bypassing the use of other compounds for the cross-linking step in hydrogels. Combined with PEI, the hydrogels were split in half and placed back with a different hydrogel that was cut. After 2 h, the split gels had “self-healed” into a singular piece of hydrogel without any cuts [73]. There was no longer a noticeable cut in the tested hydrogel. Information gathered in this study of self-healing hydrogels is crucial to the development of bone tissue grafts that can heal seamlessly in patients. The development of a “self-healing” hydrogel opens the door for implants and bone grafts implanted in the body to speed the recovery of a patient and the potential for improved quality of life post-surgery.

Many studies have endeavored to explore the applications of bioprinting in bone regeneration and repair; however, none have yet to test the scaffolds themselves in humans. Yao et al. printed bone scaffolds using polycaprolactone-hydroxyapatite (PCL-HA) powder and implanted them into rabbits with parts of their femoral and lumbar spine removed; the scaffolds were demonstrated success in supporting physiologically relevant loads. CT-based anatomical data was combined with CAD technology to produce these complex bone scaffolds [40,74]. Wang et al. printed porous scaffolds using poly-propylene fumarate resin (PPF), a degradable polymer. Scaffold degradation and changes in physical parameters, such as porosity and pore size, were observed over a 224-day period to measure any changes. Cell culture testing also demonstrated that the 3D-printed scaffolds maintained mechanical stability during their degradation and did not induce any significant cell death [40,75]. Pati et al. printed scaffolds ornamented with a bone-like extracellular matrix that mimicked the natural bony environment of the body. The printed scaffolds were shown to support osteoblastic differentiation and induced significant bone formation in rats in vivo [40,76]. A recent study by Celikkin et al. performed an in vivo study using gelMA and MSCs to heal a condyle defect labeled as critical. The study showed that gelMA and MSCs together have enhanced ability for the differentiation of the MSCs due to the porous nature of the gelMA. The MSCs that were encapsulated with the gelMA showed potential for future BTE experiments [77]. Beyond scaffolds, 3D printing can be used for a wide array of bone repair applications, including screws, plates, and gels [32]. These recent advances are critical in changing the field of 3D bioprinting of BTEs for bone.

## 5. Conclusions

The 3D bioprinting field is a relatively new field of exploration. Based on the current standard practices for bone replacement surgeries, the research conducted with 3D bioprinting can drastically impact advancements in bone healing. The developments that are being performed in the hydrogel and 3D bioprinting fields can be applied together to create a new therapy for traumatic bone injuries. With the recent developments in both fields, research is being conducted on how to apply the new advancements in hydrogels and 3D bioprinting to bone replacement and healing.

## Figures and Tables

**Figure 1 cells-12-01230-f001:**
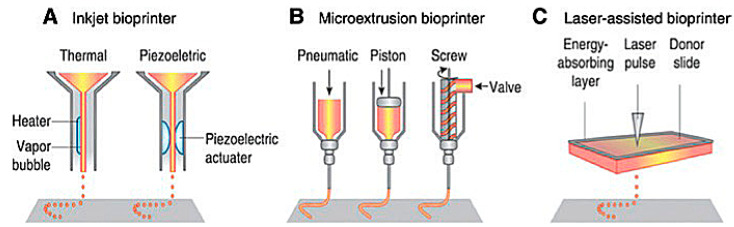
Schematic of present bioprinting techniques: (**A**) Inkjet bioprinters use a source of heat to create vapor in the print head. This vapor generates pressure within the nozzle that forces bioink from the tip. Piezoelectric print heads use pulses generated through ultrasound or piezoelectric pressure. (**B**) Extrusion printers use a mechanical dispensing system, requiring a screw, piston, or pneumatic pressure. (**C**) Laser bioprinting uses a laser pulse that strikes a point on a laser-absorbent surface; this surface generates a miniature burst of vapor that drives the biomaterial onto the substrate [6].

**Figure 2 cells-12-01230-f002:**
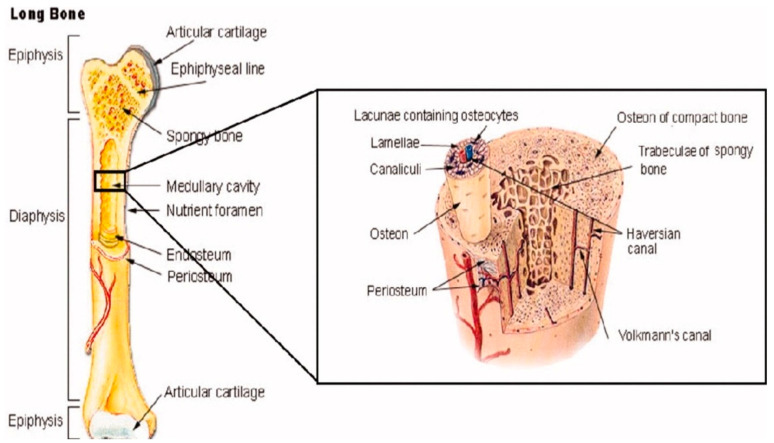
The anatomical structure of a long bone depicting basic anatomy, differences in cancellous and cortical bone, and other microstructural features taken from Porter et al. [12].

**Figure 3 cells-12-01230-f003:**
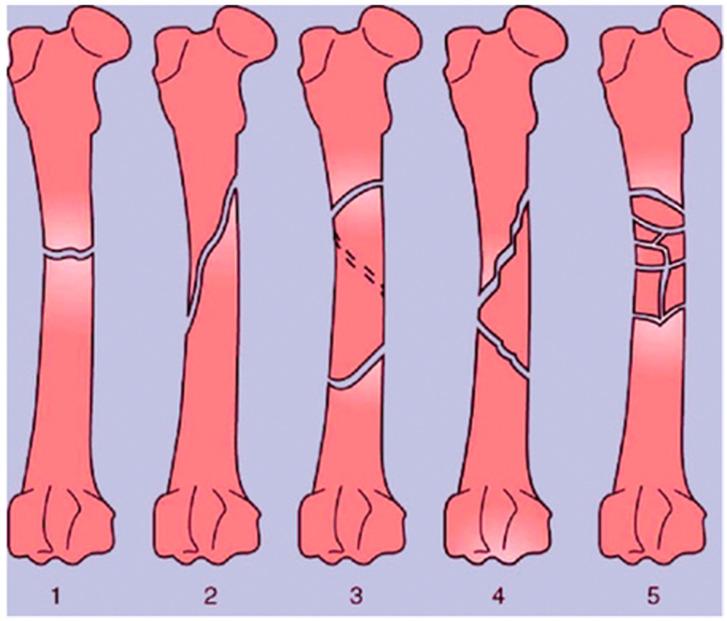
Types of long bone fractures, including transverse (**1**), oblique (**2**), spiral (**3**), comminuted (**4**), and multiple (**5**). Taken from Bigham-Sadegh et al. [16].

**Figure 4 cells-12-01230-f004:**
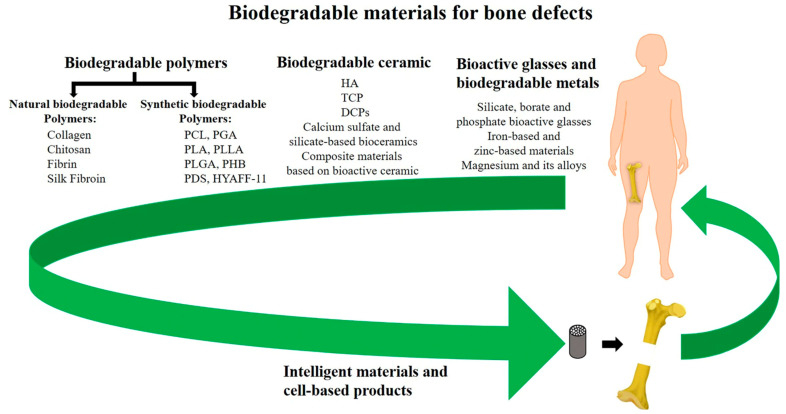
Summary of synthetic and natural biodegradable polymers and their composites that have been developed into orthopedic implants taken from Wei et al. [38].

**Figure 5 cells-12-01230-f005:**
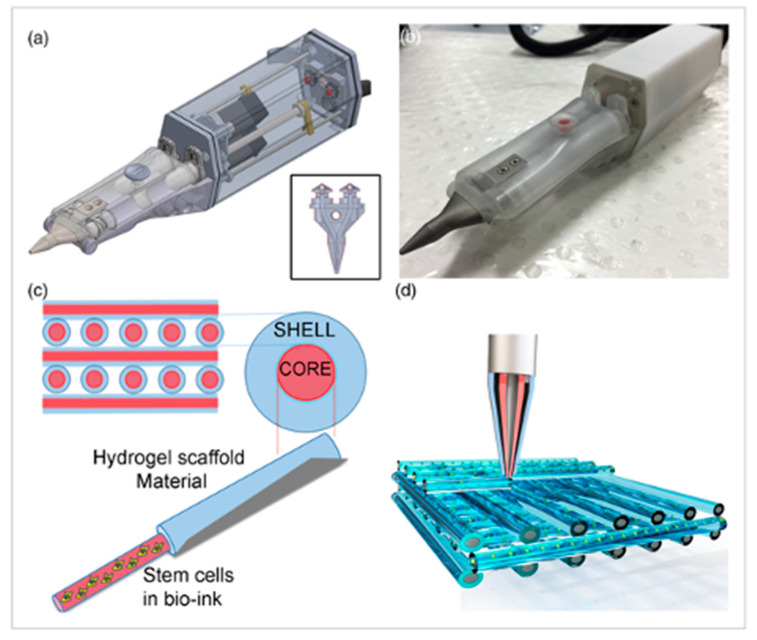
The Biopen is used for cartilage regeneration. (**a**) Schematic of the two chambers inside the Biopen and coaxial printing mechanism. (**b**) The Biopen photographed. (**c**) Coaxial printing allows for division between the shell and core. (**d**) Three-dimensional scaffolds can be printed in specific patterns using this handheld device. The schematic is taken from Di Bella, C. et al. [49].

**Table 1 cells-12-01230-t001:** Summary of recent studies that used 3D bioprinting applications for repairing bone defects, adapted from Genova T. et al. [42].

Cell Types, Molecules	Bioink	Bioprinting Modality	Application
Bone marrow MSCs, osteoblast	GelMA + nanocrystalline HA [50]	LBB (Stereolithography)	Breast cancer bone metastases
Osteoblast, breast cancer cells	PEG hydrogel + nanocrystalline HA [51] Hydrogel resins (PEG, PEG-diacryilate) [52]	LBB (Stereolithography)	Breast cancer bone metastases
Without cells	(PLA) and acrylonitrile butadiene styrene (ABS) [53]	EBB with Fused deposition model (FDM)	Radius fracture repair
Periosteal derived cells	Alginate hydrogel + collagen I, II [54,55]	EBB by piston-driven system	Periosteum Tissue Engineering
MSCs	RGD alginate hydrogels [56]	EBB by multiple-head 3D printing system	To engineer endochondral bone
ASCs	HA-GelMA [49,57]	EBB by Biopen	Regeneration of chondral lesions
Meniscal fibrochondrocytes (MFCs)	meniscus extracellular matrix (MECM)-based hydrogel [58]	3D printing fused deposition modeling	Meniscus regeneration
IPS cells, 143B human osteosarcoma cells, preosteoblasts MC3T3	Alginate hydrogel [59]	Direct- volumetric Drop-on-demand (DVDOD) technology	Microtissue fabrication and drug delivery
Simvastatin	copolymeric blend of polymers: polypropylene fumarate (PPF), PEG-PCL-PEG, and pluronic PF 127 [60]	LBB	Drug delivery
Resveratrol and strontium ranelate	PCL/hydrogel [53]	EBB	Cranio-maxillofacial regeneration

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
