# Peer review of "Three-Dimensional Bioprinting Applications for Bone Tissue Engineering"

_cells, 2023, doi:10.3390/cells12091230_

Round 1

Reviewer 1 Report

This is a well-written review of 3D bioprinting applications for bone tissue engineering. I suggest it for publication after the following minor points are addressed.

1. The resolution of Figures 1 and 2 should be improved to a higher level.

2. Line 314, alginate is not a hydrogel but one kind of polymer. 3

3. Line 318-319, alginate has relatively weaker mechanical properties. That's not correct. In most cases, alginate hydrogel has relatively weak mechanical properties.

4. 

Author Response

We thank the Reviewer for their comments and respond in detail below

  1. The resolution of Figure 1 and 2 should be improved to a higher level. We agree with this comment and have now preovided a higher resolution version of these Figures in this version.
  2. Line 314 Alginate is not a hydrogel but one kind of polymer 3. We have modified the text to clarify this point.
  3. Line 318-319 alginate has relatively weaker mechanical properties. That is not correct, in most cases alginate hydrogel has relatively weak mechanical properties. We have edited Weaker to Weak as per the Reviewers suggestion.

Reviewer 2 Report

Dear Editor,

This is an interesting title and authors reviewed the normal bone physiology and pathophysiology and how bioprinting can be adapted to further the field of bone tissue engineering. There some comments which should be considered before next steps:

-Abstract should be more and mention the novelty of current review article and what aspects are involved in your manuscript

-Introduction is poor and it needs to be more explained 

- The stem cell part and its explanation have been missed and authors should explain more about stem cell and they can use the below reference:

https://www.hindawi.com/journals/sci/2022/5304860/ (The Role of Epigenetic in Dental and Oral Regenerative Medicine by Different Types of Dental Stem Cells: A Comprehensive Overview).

-Discussion is poor

-Reference section needs recent refences (2018-2023)

Best Regards, 

Author Response

We thank the Reviewer for their comments and provide responses to each query.

Abstract should be more …longer and provide more details on what is novel in our Review. Unfortunately we cannot make the Abstract Larger as we are limited by the constrains imposed by the journal. As our article is a review of the field it is difficult to highlight what is novel since we have attempted to cover the area that was assigned to us by the editor when we agreed to write this review.

Introduction is poor and it needs to be more explained. We are not sure what further explanation we can include in the introduction. We note that in the next comment the Reviewer says we have missed discussing Stem Cells and we have included some information on Stem Cells and included the Review Article that the reviewer suggested.

Discussion is Poor. We are not sure about this comment and disagree that the Discussion is poor. We have reviewed again and tried to add comments where appropriate.

Reference section needs recent references. We thank the reviewer for these comments and have now added updated references.

Reviewer 3 Report

The authors of this manuscript provide a comprehensive review of the literature on "3D Bioprinting Applications for Bone Tissue Engineering." The paper is well-written, well-planned, and provides a meaningful selection of relevant articles. However, there are some concerns that need to be addressed.

1. While the review covers the research status of 3D bioprinting aimed at bone regeneration, similar review papers have already been published by related researchers. It is unclear what distinguishes this review from previous ones such as PMID: 36507261, PMID: 32429135, and PMID: 35480972. The authors could clarify how their review is different and what novel insights it provides.

2. The manuscript should include research such as in situ 3D bioprinting, smart-responsive materials based 3D bioprinting et al., as these areas are becoming increasingly important in the field.

3. Some of the data cited in the manuscript are outdated, with Table 1, for example, containing information from a paper published in 2011. It is recommended to update the table with more recent papers to provide a more current overview.

4. Table 2 could also benefit from more recent research being included to provide readers with a more up-to-date perspective on the subject matter.

Overall, the review is well-structured and well-informed, but some updates are needed to ensure that it remains relevant and informative.

Author Response

We thank the reviewer for their comments and appreciate that they liked our presentation . We will attempt to provide answers to the specific questions raised.

  1. What separates our new review from others in this field PMID 36507261, 32429135, and 35480972. The authors could clarify how their review is different and what novel insights it provides. We have added the suggested references and have made comments in the introduction about what aspects we are addressing in our new review
  2. The manuscript should include research such as in situ 3D bioprinting, smart-responsive materials based 3D bioprinting. We agree however based on page constraints for the review we did not include these in this review.
  3. Some of the data presented are outdated containing materials from 2011. The table should be updated. We have tried to update both the references in the paper and also in the tables that were adapted from a recent series of publications so that may be the reason some of the references are old.
  4. Table 2 could also benefit from more recent research. As discussed in Point 3 above we agree but this was taken from a recent review so that the published article used these sources. We have tried to add more recent literature to make sure that we have many of the salient points that the reviewer would like

Reviewer 4 Report

In my opinion, this review article is not well done: it is not well structured, superficial with missing information and take home message. The chapter of bone biology and fractures is oversized – although this aspect is certainly important, researchers in the field of bone tissue engineering are familiar with the bone anatomy. Similarly, the explanation of the different types of long bone fractures is only interesting if it can be related somehow to the existing 3D bioprinting techniques. The cited literature certainly does not sufficiently correspond to the current state of knowledge of 3D bioprinting.

For publishing in the Journal Cells, the topic of vitality and functionality of bone and endothelial cells before/after 3D printing should be also focussed. This is almost completely lacking here as well as the treatment of the field of mechanical issues.

In my opinion, following aspects are important and partially or fully lacking in this review article with the title: 3D Bioprinting Applications for Bone Tissue Engineering

  1. Bioprinting techniques used for 3D printing of BTE grafts
  2. Materials with their topography and porosity used for BTE bioprinting
  3. Injectable gels/materials used for application in bone defects/fractures?
  4. Assessment of (bone (MSCs/osteoclasts/endothelial) cell viability and functionality (osteogenic/angiogenic potential) after bioprinting
  5. Problems that arise with 3D printing in terms of mechanical properties. Is there a solution to this problem?
  6. Are there any recommendations for future directions in order to advance this field of research?

Author Response

We thank the Reviewer for their comments and concerns and have tried to provide an answer to each of their concerns.  We are sorry that the Reviewer does not like the approach we took and that they feel it is superficial and no take home message. As this review is for Cells we felt that not all readers will be intrically versed in bone anatomy and cell types so that was our reason to apply a detailed section on bone and fractures. Addressing the 6 points raised

1.Bioprinting techniques used for BTE. We had that in the last version and have tried to add some additional references that look at some of the new techniques that have been attempted.

2.Materials with their topography and porosity used in BTE printing. We had addressed this in the last version and not sure what else to add. As this is a cell based review rather than a materials application review we were limited by space as to what we could include.

  1. Injectable gels/materials used for application of bone defects. As this review was solicited to address only bioprinting we are not sure how injectable materials fit into this review.
  2. Assessment of cell viability and functionality after bioprinting. We did address this in our last verison

5.Problems that arise with 3D Bioprinting in terms of mechanical properties. Is there a solution. We are not sure how to address this point as it would be based on our own biases and the purpose of the invited review as to discuss all techniques that are presently available  and should not bias readers in one direction.

6.Any recommendations for future directions. We did discuss this in our final sections of the review about new and novel technologies that could potentially be tried for this field.

Round 2

Reviewer 2 Report

Dear Editor, 

The revised manuscript is acceptable.

Best Regards,